# Seasonal Changes in Vertical Distribution and Population Structure of the Dominant Hydrozoan *Aglantha digitale* in the Western Subarctic Pacific

Mari Aizawa [1], Tian Gao [1] and Atsushi Yamaguchi [1,2,*]

[1] Graduate School of Fisheries Sciences, Hokkaido University, 3–1–1 Minato-cho, Hakodate 041-8611, Hokkaido, Japan; o1417marimari@gmail.com (M.A.); gaotianbio@yeah.net (T.G.)

[2] Arctic Research Center, Hokkaido University, Kita-21 Nishi-11 Kita-ku, Sapporo 001-0021, Hokkaido, Japan

\* Correspondence: a-yama@fish.hokudai.ac.jp

**Abstract:** Hydrozoans are numerically dominant taxa in gelatinous zooplankton communities of the worldwide oceans and play an energy transfer role connecting primary producers and higher trophic level organisms. In the western subarctic Pacific, St. K2 has been established as a long-term time-series monitoring station. Various studies on zooplankton have been conducted, while hydrozoans have not been treated. This study presents the abundance, vertical distribution, and population structure of the dominant hydrozoan species (*Aglantha digitale*) at St. K2. Samples collected by vertical stratification samplings from eight layers of 0–1000 m both day and night during four seasons in one year. Hydrozoans occur throughout the year. The annual mean abundance of *A. digitale* was 198.4 ind. $m^{-2}$ and composed of 91.9% of hydrozoans. The vertical distribution of *A. digitale* was concentrated for the epipelagic layer (0–200 m), both day and night of the most season. The bell height (BH) of *A. digitale* ranged between 2.4–18.9 mm. Most of the mature individuals, with gonad length larger than 10% of BH, occurred only in July. The BH of mature individuals ranged from 4.7 to 17.6 mm, with the BH of most mature individuals were larger than >10 mm. Through observation on BH at each sampling layer, small individuals with BH < 6 mm were distributed below 300 m depths throughout the seasons, expanding their vertical distribution to the deeper layers. Inter-region comparison of abundance, maturation body size, and generation length of *A. digitale* revealed that these parameters are varied with the region and depend on the marine ecosystem structures.

**Keywords:** hydrozoa; *Aglantha digitale*; abundance; population structure; life cycle





## 1. Introduction

From worldwide oceans, hydrozoans are numerically dominant taxa in gelatinous zooplankton communities, feed on small zooplankton and serve as food for fishes, and act as mediators transporting energy between primary producers and higher trophic level organisms [1]. Recent climate changes (North Atlantic Oscillation, North Pacific Decadal Oscillation, El Nino, etc.) have had a significant impact on standing stocks of jellyfish in worldwide oceans, and may increase the duration of occurrence of hydrozoans in the temperate to subarctic regions [2]. This increase in hydrozoans is predicted to alter the balance of prey-predator relationships in marine ecosystems, with significant effects on lower and higher trophic-level organisms [3,4]. On the other hand, the chemical compositions of hydrozoans are characterized by low carbon but high nitrogen and phosphorus contents, and their outbreak affects the nutrient balance (C:N:P ratio) of the marine ecosystem, and the decomposition of their carcasses by bacteria has a significant impact on the marine mineral cycles [5,6].

In the Northern Hemisphere, *Aglantha digitale* is the most abundant hydrozoan in both abundance and biomass at high latitudes [1]. *A. digitale* has been reported as the most dominant hydrozoan species in the western subarctic Pacific [7,8], eastern subarctic

Pacific [9], northern North Pacific [10], western Arctic Ocean [11], Toyama Bay in the southern Japan Sea [12], Norwegian fjords and Svalbard Islands [13–15], eastern North Atlantic [16,17], Irish Sea [18], and White Sea [19,20]. As the ecology of *A. digitale*, feeding mechanisms [21,22], feeding impact [13,20,23], and swimming behavior [24] have been reported. For life histories, there have been reports from various oceans, including the Oyashio region [7,8], the eastern subarctic Pacific [9], Toyama Bay [12], the western North Atlantic [16], the Norwegian fjords [14], and the White Sea [19].

In the western subarctic Pacific, St. K2 has been set as a long-term time-series station to monitor various biogeochemical parameters [25]. As the information on hydrozoans at St. K2 bimodal vertical distribution with peaks at 0–50 m and 200–300 m, with a minimum at 150–200 m, and no day/night differences in abundance have been reported [26]. At St. K2, there are several studies on zooplankton [27–31], but no specific studies on hydrozoans are available. Despite their importance on the lower and higher trophic level organisms and material transport, information on hydrozoans in this area are currently scarce.

This study conducted abundance, vertical distribution, and population structure of the dominant hydrozoans (*A. digitale*) at St. K2 in the western subarctic Pacific using formalin-preserved samples collected from day and night vertically stratified samplings between 0–1000 m covering four seasons of one year. As new scientific information, seasonal and day-night changes in bell height (BH, Figure 1) and gonad maturation at each sampling depth down to 1000 m have been evaluated. The results were compared with the same species in the various regions of high-latitude waters of the Northern Hemisphere.

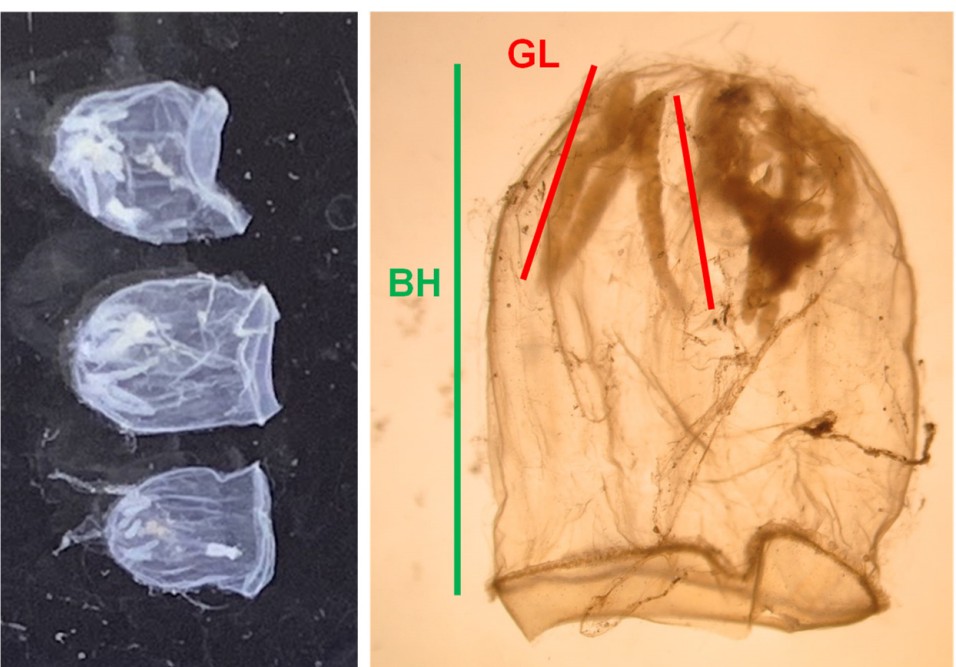

**Figure 1.** Picture on specimens of *Aglantha digitale* (**left**) and their measured body parts: bell height (BH) and gonad length (GL) (**right**).

## 2. Materials and Methods

### 2.1. Field Sampling

Day and night vertically stratified zooplankton samplings were made by oblique tow of Intelligent Operative Net Sampling System (IONESS, SEA Co., Ltd., Bristol, UK) equipped 335 μm mesh and 1.5 m$^2$ mouth area, from eight layers (0–50, 50–100, 100–150, 150–200, 200–300, 300–500, 500–750, 750–1000 m) at St. K2 (47° N, 160° E, 5230 m depth, Figure 2), located in the western subarctic Pacific on 29 October 2010, 26 February, 22–23 April, and 3–4 July 2011 (Table 1). After collection, zooplankton samples were immediately preserved with 4% buffered formalin seawater. At each sampling occasion,

environmental data such as water temperature, salinity, dissolved oxygen (DO), and chlorophyll *a* (Chl. *a*) fluorescence were measured by fluorometer and DO-sensor mounted CTD (SBE 911 plus; Sea-Bird Electronics Inc., Bellevue, WA, USA). The details of the environmental data and zooplankton biomass have been published elsewhere [27].

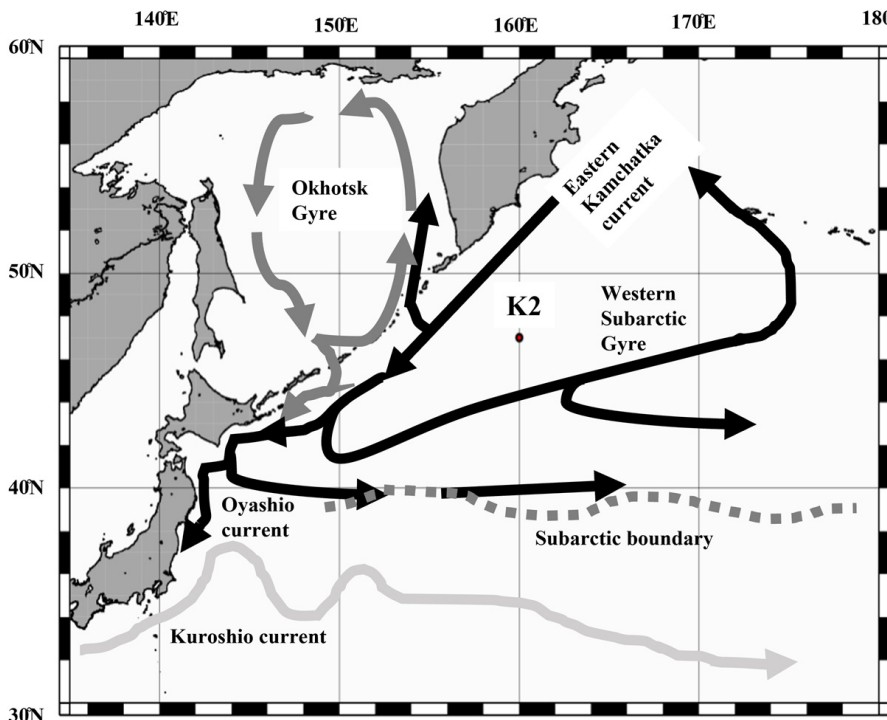

**Figure 2.** Location of sampling station K2 (47° N, 160° E) in the western subarctic Pacific. The approximate positions of the currents are superimposed [32].

**Table 1.** Zooplankton samplings (eight vertical stratification samplings between 0–1000 m) at St. K2 in the western subarctic Pacific gyre. D: day, N: night.

| Season | Sampling Date | Local Time (Day/Night) |
|--------|---------------|------------------------|
| Autumn | 29 October 2010 | 12:09–13:52 (D) |
|        | 29 October 2010 | 22:09–23:38 (N) |
| Winter | 26 February 2011 | 12:35–14:41 (D) |
|        | 26 February 2011 | 22:01–23:56 (N) |
| Spring | 22 April 2011 | 21:59–23:56 (N) |
|        | 23 April 2011 | 12:45–14:37 (D) |
| Summer | 3 July 2011 | 12:05–13:55 (D) |
|        | 3–4 July 2011 | 22:51–0:55 (N) |

*2.2. Sample Analysis*

In the land laboratory, hydrozoans were sorted from the sub-samples divided at 1/2–1/64 according to the amount of samples. Bell height (BH) and gonad length (GL) of *A. digitale*, the numerically and biomass-dominated hydrozoans, were measured at a precision of 0.05 mm by using an eyepiece micrometer under a stereomicroscope (Figure 1). Individuals with more than 10% GL in BH were treated as mature individuals [7,12,16].

As an index of the vertical distribution, the depth of the population center ($D_{50\%}$) was calculated using the following formula [30,33].

$$D_{50\%} = d_1 + d_2 \times \frac{50 - p_1}{p_2}$$

where $d_1$ is the depth (m) of the upper depth of the 50% individual occurrence layer, $d_2$ is the sampling depth interval (m) of the 50% individual occurrence layer, $p_1$ is the cumulative individual percentage (%) that occurred at depths shallower than the 50% individual occurrence layer, and $p_2$ is the individual percentage (%) at the 50% individual occurrence layer. Day-night differences in vertical distributions on each collection date were tested by the Kolmogorov-Smirnov test.

BH of *A. digitale* was expressed by histograms based on the integrated abundance (ind. $m^{-2}$) at 0–1000 m water column for the day and night of each sampling date. The depth composition of each BH interval (1 mm) was also calculated.

## 3. Results

### 3.1. Hydrography

Vertical distributions of temperature, salinity, dissolved oxygen, and fluorescence at each sampling date are shown in Figure 3. Throughout the season, the temperature was at 0.7–8.5 °C, salinity for 32.5–34.5, DO ranged between 0.6 and 10.4 mg $L^{-1}$, and fluorescence was 0.02–2.32. Seasonal thermocline developed around 50 m in July and October, and temperatures were almost constant for 100 m in February and April. For all seasons, temperature showed a minimum of 1–2 °C approximately at 100 m, then had a maximum of about 3.5 °C around 200 m, and finally decreased with increasing depth. Salinity increased with depth for all seasons. Salinity was similar for depths below 100 m in February and April, while was below 33 at <50 m, forming surface halocline in July and October. DO decreased with increasing depth and was extremely low, less than 2 mg $L^{-1}$ below 200 m depth. Fluorescence was high at the surface layer above the thermocline (<50 m) in July and October and at 0–100 m in February and April.

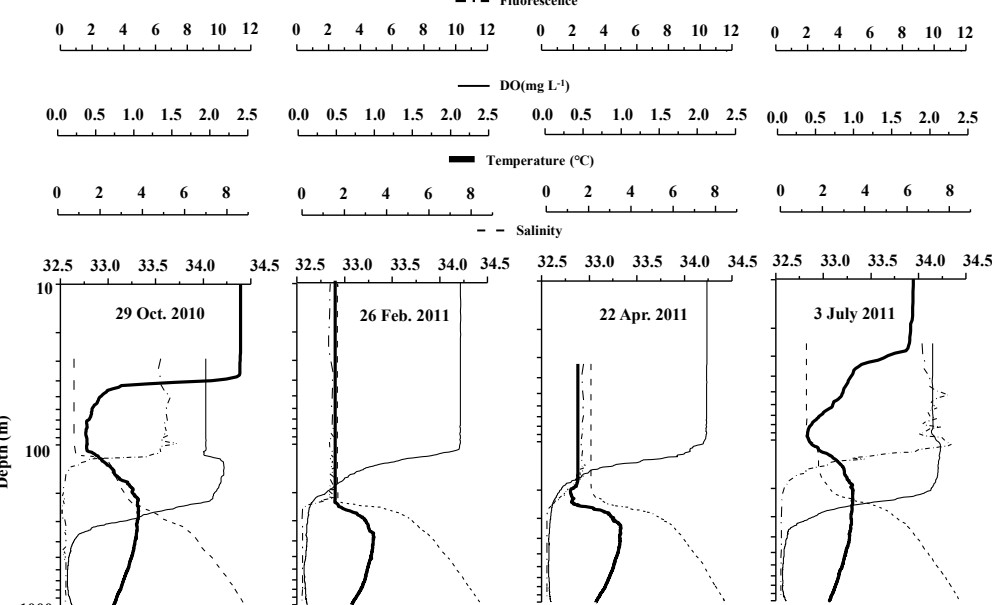

**Figure 3.** Vertical distribution of temperature, salinity, dissolved oxygen, and fluorescence at St. K2 in the subarctic Pacific gyre on four occasions from October 2010 to July 2011. Note that the vertical scale (depth) is in the logscale.

### 3.2. Aglantha digitale

Hydrozoans occurred in the western subarctic Pacific throughout the year. *A. digitale* was the most abundant species, with an annual mean abundance of 198.4 ind. $m^{-2}$ at 0–1000 m water column, composed of 91.9% of hydrozoan abundance (Table 2).

**Table 2.** Annual mean standing stocks of cnidarian species for 0–1000 m water column at St. K2 in the western subarctic Pacific from October 2010 to July 2011. Values are mean $\pm$ SD.

| Family | Species | Standing Stock | |
| --- | --- | --- | --- |
| | | (ind. m$^{-2}$) | (%) |
| Rhopalonematidae | *Aglantha digitale* Muller, 1776 | 198.4 $\pm$ 107.8 | 91.9 |
| Other Cnidaria (including fragments difficult to make species identification) | | 17.4 $\pm$ 10.1 | 8.1 |

The day-night vertical distribution and $D_{50\%}$ of *A. digitale* at each sampling date are shown in Figure 4. The vertical distribution of *A. digitale* was concentrated at 0–200 m both day and night for most seasons. While in October, the certain population density was seen to extend below 300 m both day and night. The highest population density of *A. digitale* was 1.36 ind. m$^{-3}$ at 100–150 m depth during a night in February. $D_{50\%}$ of *A. digitale* was at 26–129 m depth through day and night of all seasons. Diel change in vertical distribution was observed in July, with significantly shallower depths at night ($p < 0.05$).

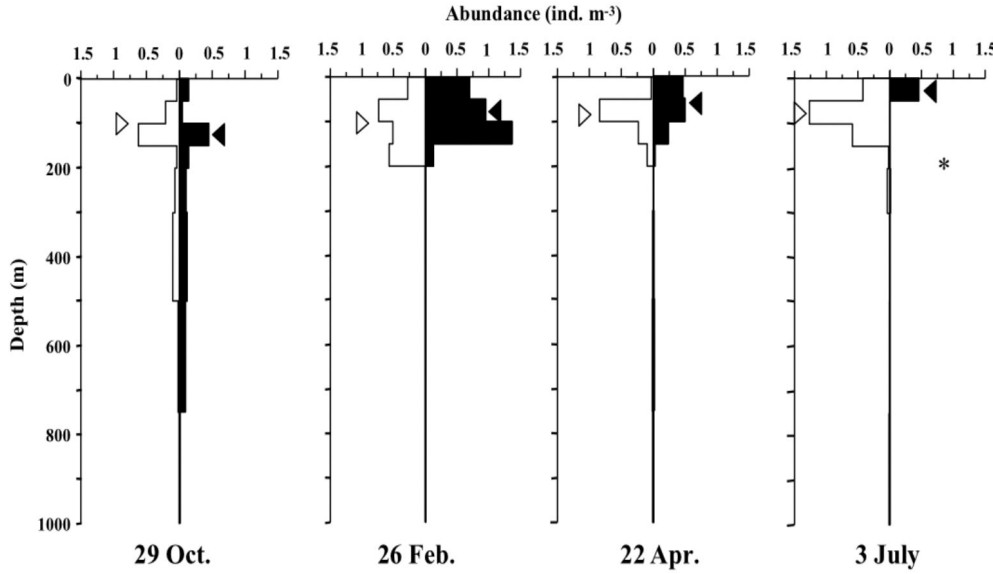

**Figure 4.** Day (open) and night (solid) vertical distribution of *Aglatha digitale* at St. K2 in the western subarctic Pacific gyre during four sampling occasions (October 2010, February, April, and July 2011). Distribution cores ($D_{50\%}$) are shown by the triangles. Diel changes were tested by the Kolmogorov-Smirnov test (*: $p < 0.05$).

Histograms on BH of *A. digitale* and their vertical distribution composition both day and night at each sampling date are shown in Figure 5. The BH of *A. digitale* ranged between 2.4 and 18.9 mm. In October and February, the most numerous individuals were seen for the BH at 8–16 mm both day and night. On the other hand, two modes of BH at 3–5 mm and 8–17 mm were observed for both day and night in April. In July, the majority of individuals was at the BH in 8–17 mm both day and night. Most of the mature specimens, GL composed >10% of the BH, occurred only in July. The BH of the mature individuals ranged from 4.7 to 17.6 mm, with most of them was at >10 mm.

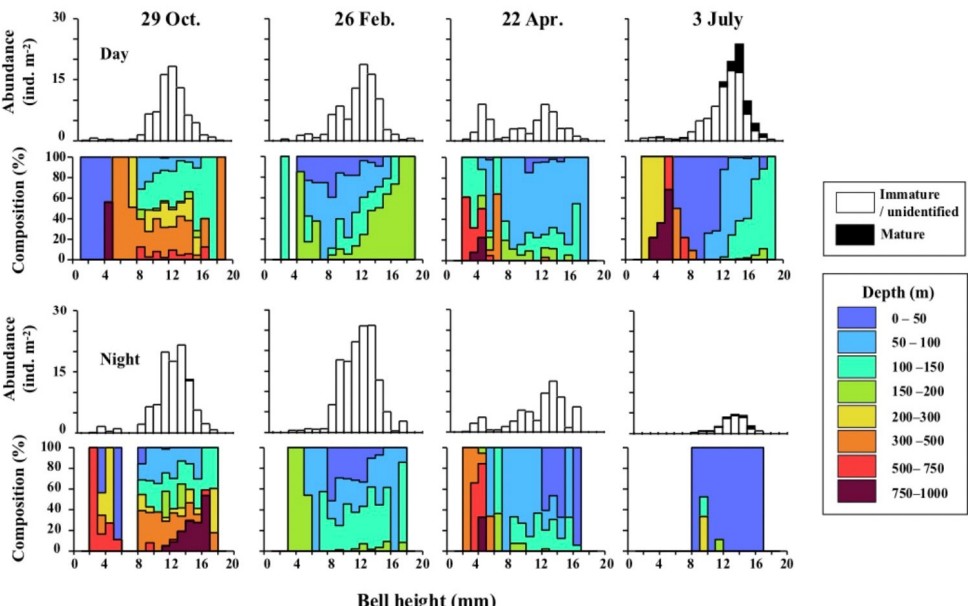

**Figure 5.** Histograms on bell heights of *Aglantha digitale* integrated over 0–1000 m water column at St. K2 in the western subarctic Pacific gyre during the day (upper) and night (lower) of the four sampling occasions (October 2010, February, April, and June 2011). Depth distribution compositions within the eight depth strata of 0–1000 m water column is also shown for each panel.

Concerning the vertical distribution composition of each BH of *A. digitale*, the distribution below 300 m was seen in October for all BH sizes, indicating that the extension of vertical distribution down to the deeper layer was a typical feature of all BH ranges (Figure 5). In February, the modal BH individuals at 8–16 mm BH were distributed near the surface both day and night, while other smaller and larger individuals were distributed at deeper layers. In April, small individuals with BH < 6 mm were distributed below 300 m, while most the individuals with a BH > 12 mm were distributed below 100 m. In July, individuals with BH smaller than 6 mm were distributed at >200 m, but at night, the main individuals with BH between 8 and 16 mm were concentrated for the surface layer at 0–50 m.

## 4. Discussion

### 4.1. Abundance of A. digitale

At St. K2, *A. digitale* was the predominant hydrozoan, accounting for more than 90% of the hydrozoan abundance (Table 2). *A. digitale* is a cosmopolitan species widely distributed at high latitude areas in the Northern Hemisphere [2,4,13]. Information on abundance, BH, BH of mature specimens, and generation time of *A. digitale* reported from various locations are summarized in Table 3. In the present study, the abundance of *A. digitale* ranged between 58.5 and 391.1 ind. m$^{-2}$ for the 0–1000 m water column. This value well corresponds with the values reported in the subarctic Pacific (368 ind. m$^{-2}$) [10] and the northeastern North Pacific (38–221 ind. m$^{-2}$) [11]. As for the marginal areas of the western subarctic Pacific, values in the Oyashio region during spring (16–316 ind. m$^{-2}$) [8] and values based on the annual sampling in the Oyashio region (55–896 ind. m$^{-2}$) [7] are also comparable. For the areas with similar values of *A. digitale* abundance in this study (<100 ind. m$^{-2}$), the Norwegian fjords (Korsfjord and Fanafjord) [14] and the Arctic Ocean [15] are available. On the other hand, high abundances of *A. digitale* over 1000 ind. m$^{-2}$ have been reported for Toyama Bay in the southern Japan Sea (maximum: 4427 ind. m$^{-2}$) [12], the Irish coast in the North Atlantic (5350 ind. m$^{-2}$) [18] and the White Sea (5000 ind. m$^{-2}$) [19] (Table 3). A characteristic of these regions is the semi-enclosed embayment having a limiting water exchange with the outer region. These facts suggest that the semi-enclosed area may have

maintained a higher population of *A. digitale* without transportation by flushing discharge caused by the ocean currents. In summary, the abundance of *A. digitale* is less than 1000 ind. $m^{-2}$ for the oceanic open area where the current transport would prevent to accumulation of high density/abundance. While in the semi-enclosed embayment condition, *A. digitale* can maintain high density/abundance (>5000 ind. $m^{-2}$) under low flushing and transport to the other region.

**Table 3.** Regional comparison on abundance, bell height, and generation length of *Aglantha digitale* from worldwide ocean.

| Region | Sampling | | Abundance (ind. $m^{-2}$) | Bell Height (mm) | | Generation Length (year$^{-1}$) | Reference |
|---|---|---|---|---|---|---|---|
| | Season | Depth (m) | | Range | Mature | | |
| Toyama Bay, southern Japan Sea | Annual | 0–500 | 73–4427 | 1–17 | 6–17 | 2 | [12] |
| Eastern/Western subarctic Pacific | Summer | 0–150 | 0–368 | – | 8.5–15.2 | – | [10] |
| Oyashio region | Annual | 0–2000 | 55–896 | 1–23 | 11–23 | 1 | [7] |
| Oyashio region | Spring | <200 | 16–316 | 4–18 | – | 1 | [8] |
| northeastern North Pacific | Summer | 0–150 | 38–221 | 0.6–17 | 1.1–16 | – | [11] |
| Northern Pacific | Summer | <200 | – | 5–20 | 15< | 1 | [9] |
| Southern Irish coastal water | Summer | 0–25 | <5350 | – | – | – | [18] |
| Northeast Atlantic Ocean | Autumn/winter | 0–100 | – | 1–18 | – | – | [17] |
| Korsfjord and Fanafjord | Annual | 0–640 | 156–358 | – | – | 2 | [14] |
| White Sea | Annual | 0–100 | 5–5000 | – | 8–12 | 1 | [19] |
| High-Arctic coastal | Annual | 0–180 | <720 | – | – | – | [15] |
| Western Subarctic Pacific (K2) | Annual | 0–1000 | 58.5–391.1 | 2.4–18.9 | 4.7–17.6 | 1 | This study |

### 4.2. Vertical Distribution of A. digitale

Vertical distribution of *A. digitale* was concentrated at <200 m for most seasons. Seasonally, diel changes in vertical distribution were seen in July when the thermocline developed, and deeper distribution (>300 m) was observed in October (Figure 5). In July, it should be noted that the day-night differences in abundance were substantially low at night and no size fraction less than 8 mm at night-histogram. It can be concluded that there is a spatial horizontal heterogeneity in the distribution of medusae. While diel changes in vertical distribution were denied in this study, nocturnal ascent diel vertical migration of *A. digitale* has also been reported from the Saanich Inlet off Vancouver [9]. The deepening vertical distribution of *A. digitale* has also been reported in the fjords of Svalbard from August to October [15]. These findings correspond with the results of this study.

As a new finding of this study, the depth distribution at each BH was determined with season and day/night. Small individuals (<6 mm BH) were distributed for the deep layers or extremely shallower depths in all seasons (Figure 5). These facts suggest that small individuals with less swimming ability could be easily transported vertically from the vertical mixing and diffusion of water masses. Swimming behaviors of six hydrozoan species, including *A. digitale,* are highly varied with species, and *A. digitale* is considered to be a jet-swimming species [24,34]. Within the three hydrozoan species applying jet-swimming, *A. digitale* is the smallest body size and has the longest jet interval in time [24]. These facts indicate that *A. digitale* is most affected by vertical and horizontal water diffusion, especially for their small-sized specimen, which has less swimming ability. These low swimming abilities of the small individuals of *A. digitale* would be difficult to stay in a stable layer because of the vulnerability of the diffusion of the water masses. It induces an extremely shallower or deeper distribution of them.

The main prey item of *A. digitale* is reported to be small copepods such as *Pseudo-calanus* [20,22,24]. For *Pseudocalanus* in the western subarctic Pacific, two species: *P. minutus* and *P. newmani* are present [35]. Within them, *P. minutus* accumulates lipids in their body and is known to have a resting (diapausing) stage at the deep layer during its life history, and their descent to deep-sea achieved summer to autumn [35,36]. Since the diapausing *P. minutus* at the deep layer contains much lipids, their nutrition values would be high, and the swimming behavior of dormant copepods is reduced to maintain a low metabolic rate [37,38]. These characteristics of the deep-sea dormant *P. minutus* (small body size, high nutritional value, and low swimming behavior) suggest that they are a sufficient prey item for *A. digitale*. Thus, the extension of the vertical distribution of *A. digitale* to the deep layer during October can be explained from viewpoint of their food availability (e.g., to capture nutrient-rich *P. minutus* distributing deep layer during these seasons).

### 4.3. Population Structure and Body Size of A. digitale

The BH range of *A. digitale* observed in this study (2.4–18.9 mm) well corresponds to those reported in Toyama Bay in the southern Japan Sea [12], in the spring Oyashio region [8], in the northeastern North Pacific [11], and in the northeastern North Atlantic [17] (Table 3). It should be noted that most of the listed studies applied similar mesh sizes to this study (200–335 μm); thus, differences in the applied mesh size would be negligible. Concerning BH, large BH (up to 20 mm) have been reported from the Saanich Inlet off Vancouver and annual observations in the Oyashio region [7,9]. For such a subarctic coastal area, the abundance of small copepods, the primary prey of *A. digitale*, has been reported to be high [39]. These facts suggest that favorable food conditions would be an important factor in achieving large body sizes of *A. digitale*.

The minimum maturation size of *A. digitale*: 6 mm BH has been reported in Toyama Bay, southern Japan Sea [12]. Such a small mature specimen of *A. digitale* (8–12 mm BH) has also been reported in the White Sea [19]. Common characteristics of these areas with small maturation sizes of *A. digitale* are that these areas are semi-enclosed seas having high abundances of them (>1000 ind. m$^{-2}$) (Table 3). The feeding mode of *A. digitale* is basically carnivorous [13,20,23,24]. The energy transfer for such a higher trophic level organism is considered to be higher in the areas where the marine ecosystems are composed of simply limited species [40,41]. The regions where high abundance and small maturation body size of *A. digitale* Toyama Bay and the White Sea are characterized by semi-enclosed seas, having such high energy transfer efficiency food web and marine ecosystem structures composed by the limited species [12,19]. Thus, in the areas where the marine ecosystem is composed of the simple species structure and high energy transfer efficiency to the higher trophic level organisms, abundance of *A. digitale* would be high, and they may mature at smaller body-sizes. In this study, the observed minimum maturation size of *A. digitale* was small as 4.7 mm, while most of the mature specimens were at BH over 10 mm (Figure 5). These facts suggest that the few small body-sized mature individuals are considered to be transported to the oceanic St. K2 from the marginal areas, such as the Aleutian Islands or the Okhotsk Sea.

In this study, one or two cohorts were identified for BH of *A. digitale* on each sampling date (Figure 5). while because of the scarce sampling occasions (four times per year), it was difficult to trace the growth of each cohort. For the developmental stages, mature specimens only occurred in July (Figure 5). These facts suggest that the life history of *A. digitale* is a one-year generation length having reproduction in summer for the western subarctic Pacific. Concerning the life span of *A. digitale,* one or two generations per year have been reported in the Oyashio region [7,8], Toyama Bay in the southern Japan Sea [12], Saanich Inlet off Vancouver [9], Norwegian fjords [14], and the White Sea [19] (Table 3). Thus, the generation time in this study, one year, is an ordinary generation length for *A. digitale* in the open oceanic region.

On the other hand, two generations per year of *A. digitale* have been reported for Toyama Bay in the southern Japan Sea and Norwegian fjords (Table 3). These regions are

semi-enclosed seas where the energy transfer efficiency to the higher trophic level organisms is expected to be high. Thus, the abundance, maturation body size, and generation length of *A. digitale* would be related and determined by the component organisms in the marine ecosystem in each region. In semi-enclosed seas composed of the simple species group, because of the high energy transfer to hydrozoans *A. digitale*, they may have high abundance and small maturation sizes, which implies a shorter generation time. Thus, two generations per year of *A. digitale* have seen such areas [12,24]. The major reproductive period summer observed in this study is also consistent with the life history of this species in the adjacent Oyashio region [7]. In these regions, summer is the season when copepods, the main prey of *A. digitale,* reaches annual peak abundance at the surface layer [35,39]. It is interpreted that *A. digitale* may mature and have reproduction under sufficient food conditions during the summer season.

**Author Contributions:** Conceptualization, M.A. and A.Y.; methodology, M.A. and A.Y.; software, M.A.; validation, M.A. and A.Y.; formal analysis, M.A.; investigation, M.A.; resources, M.A.; data curation, A.Y.; writing—original draft preparation, T.G.; writing—review and editing, A.Y.; visualization, T.G.; supervision, A.Y.; project administration, A.Y.; funding acquisition, A.Y. All authors have read and agreed to the published version of the manuscript.

**Funding:** Part of this study was supported by Grant-in-Aid for Challenging Research (Pioneering) 20K20573, Scientific Research 22H00374 (A), 20H03054 (B), 19H03037 (B), and 17H01483 (A) from the Japan Society for the Promotion of Science (JSPS). This study was partially supported by the Arctic Challenge for Sustainability II (ArCS II; JPMXD1420318865) and the Environmental Research and Technology Development Fund (JPMEERF20214002) of the Environmental Restoration and Conservation Agency of Japan.

**Institutional Review Board Statement:** Not applicable.

**Informed Consent Statement:** Not applicable.

**Data Availability Statement:** Data will be available from request to the corresponding author.

**Acknowledgments:** We thank the captains, officers, crews, and researchers onboard the R/V *Mirai*, JAMSTEC, for their great efforts during the field sampling.

**Conflicts of Interest:** The authors declare no conflict of interest.

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
