# Peer review of "Seasonal Changes in Vertical Distribution and Population Structure of the Dominant Hydrozoan Aglantha digitale in the Western Subarctic Pacific"

_2673-1924, doi:10.3390/oceans4030017_

Round 1

Reviewer 1 Report

The work is interesting and overall well-done. The manuscript is clear, relevant for the field and structured mostly appropriately. It includes present-day, adequate and sufficient amount of cited literature covering all main questions.

My main comments relate to the vertical migrations of medusae and estimates of the abundance of medusae.

The authors state « The day-night differences in vertical distribution were present in July, with significant shallower distribution at night» (lines 20-21) and «Seasonally, nocturnal ascent DVM was seen in July when thermocline developed» (lines 201-202). According to Fig. 4, there are significant day-night differences in overall abundances of medusae (the integrated abundance (ind. m-2) at 0–1000 m water column) in July. There is no size fraction less than 8 mm at night-histogram. It can be concluded that there is a spatial horizontal heterogeneity in the distribution of medusae. It is hardly possible to draw such a conclusion about vertical migrations on the basis of a single station, without taking into account the spatial heterogeneity of the distribution of medusae.

The authors state "The BH of A. digitale ranged between 2.4 and 18.9 mm." (lines 140-141). It can be assumed that the authors of the study did not find the earliest stages of development, 1-2 mm in size, which can be very numerous. Apparently, this is due to the rarity of sampling. This fact should be taken into account when comparing the results with the data of other authors. The number of the earliest stages can be very high, but then abundance quickly decreases several times during the growth of medusae. You state «… the semi-enclosed area may have maintained a higher population of A. digitale without transportation by flushing discharge caused by the ocean currents» (lines 189-190). However, the higher number of medusae may also be explained by the fact that the researchers took into account the earliest stages of development (medusae less than 2 mm). In addition, the difference in the number of medusae may be due to the difference in the depth of the stations. In shallow waters, medusae can concentrate in a narrow surface layer of water.

 In addition, there are a few small comments. Please find my comments below.

The two statements contradict each other: Lines 18-19 «The vertical distribution of A. digitale was concentrated for the epipelagic layer (0–200 m), both day and night of the most season» and lines 26-28 «…small individuals with BH < 6 mm were distributed below 300 m depths throughout the seasons, expanding their vertical distribution to the deeper layers».

 Figure 2: It is difficult to understand which X-scale refers to which parameter, especially fluorescence and DO. You need to swap the headers to the X-axes for fluorescence and DO.

 Table 2: The use of the "Standard Error, SE" is justified only in the case of the normal distribution. In other cases, it is better to use the "Standard Deviation, SD".

 Table 3: Correct the link to the article Pertsova et al. (2006). Correct the depth for the White Sea to "0-100".

 Lines 201-202. Although the term "DVM" is widely used, it is necessary to provide an abbreviation meaning in the manuscript "DVM = Diel Vertical Migrations”.

 Lines 216-217: I cannot understand, why “low swimming abilities of the small individuals of A. digitale may induce an extremely shallower or deeper distribution of them”. I think many planktonic organisms have low swimming abilities. Please explain your statement.

 Line 244: «The feeding made of A. digitale is basically carnivorous». The sentence is not very clear, reformulate. May be you mean «feeding mode»?

 Line 284. «resources, M.K.», may be change to « resources, M.A.»?

 Lines 337-338. Correct the reference to «Pertsova, N.M.; Kosobokova, K.N.; Prudkovsky, A.A. Population size structure, spatial distribution, and life cycle of the hydromedusa Aglantha digitale (O.F. Müller, 1766) in the White Sea. Oceanology. 2006, 46, 228–237.»

Minor editing of English language required

Author Response

Thank you for providing valuable comments on our manuscript. 

Your comment is correct for the implication of the diel vertical migration of A. digitale in July.  We denied diel vertical migration in the revised manuscript and interpreted it due to horizontal spatial heterogeneity (L201–204). 

The effect of no collection of the small-size fraction (<2.4 mm) of this study would be due to the mesh size (335 µm) of this study.  While such an effect (no collection of the smaller-size class) would affect the regional comparison in abundance, most of the studies listed in Table 3 applied similar mesh sizes (e.g., 200-335 µm).  So we concluded there are fewer effects of the differences in used mesh sizes.  We shortly added such a note in the revised manuscript (L240–241).

For the contradicting statement of vertical distribution in the Abstract, we deleted the mention of deeper distribution because it was minor (L18–19). 

For Figure 2, OK, we placed the headers for two X-axes (fluorescence and DO) (L122–123).

For Table 2, we changed to SD (L157–158). 

For Table 3, we corrected two pieces of information on Pertsova et al. (2006) (L195–196). 

For the abbreviation of DVM, we full-spelled it in the revised manuscript (L205–206). 

For the swimming ability issue, we think that small individuals would be difficult to stay in a stable layer because of the vulnerability of the diffusion of the water masses.  In the revised manuscript, we added such remarks to a limited extent (L219–221). 

For Line 244, it was our mistake.  We corrected it as “feeding mode” (L251–252).

For M.K., we corrected it to “M.A.” (L291).

For reference of Pertsova et al. (2006), we corrected (L344–345).  Thank you for kindly pointing this out. 

Reviewer 2 Report

Dear Authors,

This manuscript has limited usefulness and interests for the greater audience in its present form. Introduction is scattered, introduction of the hydrozoan is limited, methods are not fully explained, abbreviations mostly not explained. There is no summary for the paper, general accuracy is missing. The English suffers from serious problems.

I have indicated many conceptual problems, style and linguistic problems in the text. Although I cannot recommend it to publish in its present form, but after significant improvement I am ready to reconsider.

Author Response

Thank you for providing valuable comments on our manuscript.  Before re-submitting, we received an English edit from the native speaker of our colleagues. 

Round 2

Reviewer 1 Report

The authors addressed the major issues satisfactorily. Manuscript can be accepted in present form.